# Challenging the Existing Model of the Hexameric HIV-1 Gag Lattice and MA Shell Superstructure: Implications for Viral Entry

**DOI:** 10.3390/v13081515

**Published:** 2021-07-31

**Authors:** Joy Ramielle L. Santos, Weijie Sun, Tarana A. Mangukia, Eduardo Reyes-Serratos, Marcelo Marcet-Palacios

**Affiliations:** 1Department of Medicine, Alberta Respiratory Centre, University of Alberta, Edmonton, AB T6G 2S2, Canada; joyramie@ualberta.ca (J.R.L.S.); weijie2@ualberta.ca (W.S.); mangukia@ualberta.ca (T.A.M.); reyesser@ualberta.ca (E.R.-S.); 2Laboratory Research and Biotechnology, Department of Biological Sciences Technology, Northern Alberta Institute of Technology, Edmonton, AB T5G 2R1, Canada

**Keywords:** HIV-1, matrix shell, viral entry, hexameric matrix trimers, HIV-1 structure, hosohedron, MA lune, envelope cytoplasmic domain, Env CT, fusion hub

## Abstract

Despite type 1 human immunodeficiency virus (HIV-1) being discovered in the early 1980s, significant knowledge gaps remain in our understanding of the superstructure of the HIV-1 matrix (MA) shell. Current viral assembly models assume that the MA shell originates via recruitment of group-specific antigen (Gag) polyproteins into a hexagonal lattice but fails to resolve and explain lattice overlapping that occurs when the membrane is folded into a spherical/ellipsoidal shape. It further fails to address how the shell recruits, interacts with and encompasses the viral spike envelope (Env) glycoproteins. These Env glycoproteins are crucial as they facilitate viral entry by interacting with receptors and coreceptors located on T-cells. In our previous publication, we proposed a six-lune hosohedral structure, snowflake-like model for the MA shell of HIV-1. In this article, we improve upon the six-lune hosohedral structure by incorporating into our algorithm the recruitment of complete Env glycoproteins. We generated the Env glycoprotein assembly using a combination of predetermined Env glycoprotein domains from X-ray crystallography, nuclear magnetic resonance (NMR), cryoelectron tomography, and three-dimensional prediction tools. Our novel MA shell model comprises 1028 MA trimers and 14 Env glycoproteins. Our model demonstrates the movement of Env glycoproteins in the interlunar spaces, with effective clustering at the fusion hub, where multiple Env complexes bind to T-cell receptors during the process of viral entry. Elucidating the HIV-1 MA shell structure and its interaction with the Env glycoproteins is a key step toward understanding the mechanism of HIV-1 entry.

## 1. Introduction

The outer surface of HIV-1 is surrounded by a lipid bilayer containing the viral spike envelope (Env) glycoproteins. The Env glycoproteins play a critical role in viral entry and have been extensively studied. Env glycoproteins originate from a larger glycoprotein-160 (gp160) precursor produced in the rough endoplasmic reticulum as homotrimers, later cleaved by intracellular proteases to generate the gp120 trimers (gp120) associated with the transmembrane gp41 polypeptides [1]. On average, 14 copies of the mature Env glycoprotein are found per viral particle, clustered together in patterns that resemble underlying “trenches” created by internal structural components of the virus [2]. Inside the virus, embedded into the inner lipid layer, matrix (MA) proteins arrange themselves into a shell with a superstructure that has not been elucidated. In vitro, MA have been resolved into trimers by X-ray crystallography [3]. These trimers have been observed to form hexagonal arrangements [4] when placed onto flat two-dimensional surfaces. The MA trimers themselves originate from a larger precursor group-specific antigen (Gag) polyprotein, which is responsible for viral assembly. The current model for viral assembly involves the recruitment of Gag polyproteins to the inner cell membrane. It is suggested that Gag proteins form a hexagonal lattice that is postulated to trigger membrane bending and later scission to release the newly formed immature particle [5]. Recent studies have also suggested that the hexagonal Gag lattice may be formed through the recruitment of Gag dimers or trimers [6]. In the viral maturation process, the Gag polyprotein is separated into its polypeptide derivatives, including MA, capsid (CA), nucleocapsid, spacer peptides, and p6 domains [7]. The CA proteins then assemble into the HIV-1 core. Altogether, Env, MA, and CA form the principal structural components of a mature HIV-1 virion.

All observed mechanisms of HIV-1 entry into a host cell require a critical membrane fusion step. Two pathways have been proposed to date. One involves an endocytosis step that results in viral fusion from within endosomes [8], while the second mechanism is a more widely accepted receptor-mediated fusion of viral particles to the host’s outer membrane. The initial step is triggered by the interaction of gp120 with T-cell surface glycoprotein CD4 receptors (CD4) and coreceptors CXCR4 and CCR5. Stoichiometrically, a gp120 trimer could bind a total of three coreceptors, one per gp120 subunit. However, interactions can be effective even when a gp120 trimer binds with a single CD4 receptor and coreceptor [9]. Although interactions of three CD4 receptors, each binding to one of the three gp120 in a single spike trimer, is plausible, this one-to-one stoichiometry has been considered less likely for reasons reviewed elsewhere [10].

Multiple complexes of Env glycoproteins bound to CD4 receptors are required for viral entry [11,12]. Upon binding of two or more gp120 trimers to CD4 receptor complexes on the host membrane, gp41 undergoes a conformational change by which its N-terminal fusion peptide inserts itself into the target cell membrane, inducing the gp41-mediated membrane stalk formation prior to fusion. An individual viral particle carries on average 14 gp120 trimers [2], and these trimers move within the viral membrane plane to facilitate docking of multiple gp120-CD4 interactions [13]. Interestingly, Env glycoproteins do not appear to be randomly distributed throughout the outer viral membrane but cluster in groups, suggesting that gaps exist between MA proteins to accommodate the unusually large gp41 cytoplasmic (CT) domains.

Computational, microscopic, and biomolecular techniques have contributed to an increasingly more detailed understanding of the viral entry process. However, important unanswered questions remain. One such question relates to the role the MA shell plays in viral entry. Presently, the precise arrangement of the HIV-1 MA shell remains unknown. In a recent publication [14], we examined the geometrical restrictions presented by Gag and MA trimers in the existing model for the superstructure of the shell, which involves hexagonally arranged trimers. The hexagonal arrangement, first observed by Alfadhli et al. [4], presented compelling electron microscopy images of myristoylated-MA or Gag trimers assembled into a flat two-dimensional arrangement. The HIV-1 MA shell is thought to emerge from an initial superstructure of hexameric Gag trimers, a model that warrants a careful re-examination. Previously, we analyzed the geometrical limitations of the hexameric Gag lattice and demonstrated that this arrangement is not mathematically feasible [14]. We proposed a new MA shell model consisting of a six-lune MA hosohedron that explained a number of microscopy observations, including HIV-1 viral expansion during entry [15], rearrangement of Env glycoproteins to form clusters during maturation [16,17,18], the pleomorphic nature of viral structures including spherical and ellipsoidal shapes [19], and the correct number of MA trimers per viral particle [20].

Our paper mentioned certain aspects of the HIV structure that we did not address in the proposed hosohedron model. One self-critique we pointed out was that the model started from a single MA trimer and added trimers without taking into account the incorporation of Env glycoproteins. The aim of this current study is to analyze the impact of Env glycoproteins on MA shell assembly and how our six-lune MA hosohedron model allows for Env glycoprotein clustering during viral maturation. We once again challenge the hexagonal Gag lattice arrangement, not only for its geometrical limitations but also because this rigid configuration does not allow the movement of Env glycoproteins required for viral entry. Structure and function are highly interrelated. We propose a novel interaction between the CT domains of Env glycoproteins and the MA trimers, which provides a mechanism of HIV viral maturation. In addition, the interlunar spaces in our model guide Env clustering toward the region where all interlunar gaps converge. This region of convergence, which we will designate “fusion hub”, allows sufficient space for multiple complexes of Env glycoproteins to bind CD4 receptors, which is required for viral entry. In this work, we build a structurally complete all-atom viral structural model of HIV-1 by integrating microscopy, bioinformatics, and experimental data.

## 2. Materials and Methods

Construction of hexagon tile MA lattices: The hexagonal tiles, sphere, and ellipsoid objects were constructed in Autodesk Inventor 2019. In “Part” mode, a simple hexagonal outline was centred vertically and horizontally on the Y-Z plane and extruded so that the tile would have an arbitrary thickness. On the X-Y plane cutting through the centre of the extruded tile, a reference point was created at an arbitrary sphere radius from the top face of the tile. In “Assembly” mode, a central, primary tile was identified and grounded in the program, and the second layer of tiles was inserted systematically around the primary. Each subsequent layer was constructed by adding tiles outwards from and around the primary tile and constrained to the centre, using the reference point created in “Part” mode. The ellipsoid was constructed in a similar manner, constraining each tile tangentially to the face of a hidden ellipsoid instead of the central reference point. Each completed assembly was analyzed for interference under the “Inspect” tab of the program. The interference function automatically highlighted overlapping areas in red, as shown in Figure 1 and Appendix A.

Modelling the Env glycoprotein (gp120/gp41): The crystal structure of the external domains of the Env glycoproteins, including gp120 and gp41 (PDB ID: 4ZMJ) [21], was combined with the transmembrane domain (PDB ID: 6E8W) of gp41, determined by NMR [22]. Residues L^660^-D^664^ were removed from 6E8W, as these amino acids were present in 4ZMJ. The amphipathic lentiviral lytic peptide (LLP) domains of gp41 were obtained from the NMR structure (PDB ID: 5VWL) previously determined [23]. The structure of the remaining 40 amino acid segment of the gp41 CT domain (G^200^-G^240^) that constitutes the Kennedy sequence (KS) domain was predicted using PEP-FOLD3 [24] and RaptorX [25]. We were unable to use the KS domain published by Piai et al. [26] as their structure was missing a significant portion of the KS domain. All 3D models were validated with ProSA [27] and ERRAT [28]. ProSA analysis generated z-scores of −2.24 and −2.19 for models generated with PEP-FOLD3 and RaptorX, respectively. ERRAT analysis generated results that agreed with those by ProSA. Overall quality factors for our models were 55.3 and 14.21 for PEP-FOLD3 and RaptorX, respectively. Therefore, we used the model generated by ProSA. The final structure was validated by fitting the model in a cryoelectron tomography structure (EMDB ID: EMD-1814) using PowerFit [29]. The best-fit solution had a cross-correlation score of 0.730, a fisher z-score of 0.928, a z-score/σ of 36.3 and a sigma difference of 0.00. All distances shown in figures were calculated within PyMol (Molecular Graphics System, Version 1.7.6.0, Schrödinger, LLC).

Scripts and calculations: All scripts and calculations were developed in the python language as previously described [14]. Coordinates were exported to PyMol for the construction of structures. Angstroms (Å) units were reported in nanometers (nm) for consistency. All scripts and calculations are available for download (https://sites.ualberta.ca/~marcelo/HIV-1_MA_Builder, accessed on 25 July 2021).

Mathematical algorithm used to generate coordinates for vectors to assemble the HIV-1 MA shell: We followed similar algorithms as previously described [14], with some modifications. To achieve the incorporation of Env glycoproteins in the structure in random locations, MA trimer incorporations were terminated with a probability of 1.36% (14 stops out of 1028 possible locations). The location of terminated chains was tracked throughout the script. The final MA shell model was then modified by deletion of termination point MA trimeric units and insertion of Env glycoproteins at these locations. All models were generated to contain 14 Env glycoproteins following previous observations [2].

*Interlunar distance measurements: A script* was developed to identify MA trimers at the edge of each lune. Models were opened in PyMol, and the screen was centred at the centre of the mass of edge MA trimers. The field of view variable was set at 0.01 to flatten the perspective. A total of 10 images were parsed to generate a flat map of interlunar spaces. A vector line was plotted connecting edge trimers and a tracing of the edge of the lune. The CT domain of Env glycoproteins was shown in their respective locations as determined by our random process described above.

## 3. Results

Hexagonal tiles cannot represent the MA shell superstructure: Electron tomographic evidence suggests that either Gag polyprotein trimers or MA trimers can arrange themselves into hexagonal configurations following the curvatures of spherical or ellipsoidal viral particles. Numerous reports attempted to explain microscopic observations by building models composed of hexagon tiles. Initially, we used this model-building approach to attempt constructing spherical and ellipsoidal structures. We used three hexagons and arranged them into a flat two-dimensional configuration (Figure 1Ai). Because all tiles are regular hexagons, all angles are 120°, including internal and external angles. However, when three hexagon tiles are arranged into a gently curved configuration, this is impossible without splitting the converging point between all three tiles (Figure 1Aiii). In addition to breaking the structure, this also changed the external angles to less than 120°, meaning that the addition of a new regular hexagon tile would not be possible without allowing for tile overlap. We found that construction of tile-based models of spherical and ellipsoidal shapes could not be assembled without gaps (Figure 1B,C, black arrowheads) and overlapping tiles (Figure 1B,C, white arrowheads). Ellipsoidal shapes were more affected than spherical shapes because of the more drastic change in angles starting from the initial tile (Figure 1B,C, yellow tile). In a previously published Appendix A [14], we showed that regular hexagons could not be used to build the HIV model unless one of the hexagon sides had a shorter side length.

The use of “gaps” to build MA lattices containing breaks that presumably accommodate for geometrical restrictions is not a feasible solution (Appendix A). Modelling such structures reveals that these arrangements generate lattices that force MA trimers to occupy the same space at the same time, resulting in physical overlap. Out of the 128 MA hexagons in the structure model in Appendix A, only 16 were free of overlap. The degree of overlap (red shaded areas) was easier to visualize using green hexagon tiles, but not when using triangles that represented MA trimers (Appendix A).

To visualize the difficulty of fitting hexagons into a curved surface, we built a model consisting of a paper regular hexagon attached to the surface of a balloon (Appendix A). When hexagon sides 1–2, 2–3, 3–4, 4–5, and 5–6 were secured up against the surface of the balloon, the last hexagon side 1–6 lifted off the balloon surface, creating a bulge. As a result, the length of side 1–6 was significantly shorter than all other regular sides. If each corner of this hexagon was representing an MA trimer, the trimers on positions 1 and 6 would be pushed together by the curvature of the sphere. MA trimers would counteract this compressing force by bending the surface of the sphere back to flat. Our model solves this constraint by eliminating the sixth MA trimers in each hexagon, thus allowing for incomplete hexagons. Other solutions, including hexagon cylinder configurations (Appendix A) or distorting the curvature of the viral membrane with flat patches, were tested (data not shown) and did not provide feasible solutions.

Another critical consideration was to assume that the Gag lattice forms from a single initial Gag trimer. We made this assumption based on three reasons. Firstly, we estimate a 1 to 1000 ratio of formed lattices to single Gag trimers. This assumption is based on the fact that we have roughly 1000 MA trimers in a complete model of an HIV-1 particle. Secondly, single Gag trimers have more degrees of freedom relative to the inner leaflet and move at higher speeds than bulkier pre-formed lattices. A single Gag trimer therefore outcompetes two adjacent forming lattices from docking with each other and are likelier to attach to the edges of each forming lattice one at a time. Thirdly, the recruitment of subsequent Gag trimers results in further curvature of the plasma membrane, creating a more noticeable “arc.” Many of these three-dimensional arc shapes can form simultaneously, but two adjacent arcs are unlikely to fuse because their opposing ends are positioned at a relatively obtuse angle and therefore thermodynamically unfavourable to align.

In our alternative model to hexagon tiles, we generated an MA trimer shell in which MA trimers connect to form a six-lune hosohedron (Figure 2A). A flattened representation of the shape was built to illustrate the post-entry conformation of the MA shell using MA trimers (Figure 2B) and as a superstructure silhouette to facilitate comparisons with other possible arrangements (Figure 2C). Our original model, proposed in 2019, was able to accommodate spherical and elliptical shapes but did not account for the incorporation of other proteins in the cytoplasmic side of new virions (Figure 2A). The flattened version of the structure shows a six-lune hosohedron that resembles a snowflake (Figure 2B). All six lunes contain a central core of trimers that we coloured in black to highlight the continuity of trimers from the central trimer in yellow all the way to the end of each lune. A silhouette of this structure shows small variations between the six lunes.

To enhance our previous model to more accurately depict the HIV structure, we had to incorporate the Env glycoproteins into the model in a mathematically feasible way. We started by modifying the script to randomly interrupt the incorporation of MA trimers during the formation of the shell to generate gaps where Env glycoproteins would be incorporated. Interruptions were indicated with a red sphere from which no additional MA trimers could be added. Adjacent trimer chains extended to fill the void (Figure 2D). A total of 14 interruptions were introduced at random to allow a gap for later incorporation of an Env glycoprotein. Because a total of 1028 trimers are needed to build a sphere, and 255 trimers are found along the core of each lune, there was a 24.8% probability that a random interruption would take place at a core trimer (black trimers). The majority of the randomly generated interruptions (Figure 2E) resulted in lune shapes similar to structures with no interruptions (Figure 2C), which we will designate “minimally disrupted hosohedrons”. When a core trimer was interrupted late in the lune, the core simply continued its growth around this point (Figure 2E, arrowhead).

However, the random interruptions would occasionally result in a more amorphic shape, departing from the classic six-lune hosohedron in our model. From a total of 50 spheres, each with 14 random interruptions, we showcase one sphere in Figure 2G,H with the most affected shape, which we will term “highly disrupted hosohedrons”. By chance, an early interruption of two core trimers (Figure 2H, arrowheads) generated a shape with smaller lunes (Figure 2I). Interestingly, the introduction of 14 interruptions per particle to simulate the incorporation of Env glycoproteins resulted in larger gaps between adjacent lunes (Figure 2G, arrowhead) compared with structures lacking interruptions (Figure 2B). Because Env and MA proteins are targeted to the host membrane through independent pathways, we expect random incorporation of Env glycoproteins in the Gag lattice and subsequently in the MA shell. All subsequent analyses were conducted using the structure shown in Figure 2G. The variability introduced by this random interaction drastically changes the MA shell arrangement and may play a functional role in viral entry.

Reconstruction of an Env glycoprotein: The structure of the HIV-1 Env complex has not been resolved. Available models have only assessed portions of the Env complex. We used previously established domains to reconstruct a full Env structure. We fitted the crystal structure of gp120 and gp41 [21] to the NMR transmembrane domain (TM) using an electron tomograph volume (Figure 3A,B) [22]. The Kennedy sequence (KS) domain of Env was predicted as described in the materials and methods section and attached to the NMR structure of the LLP amphiphilic helixes. The overall structure was 19.27 nm in height, with the KS domain 5.64 nm wide (Figure 3C).

Fitting Env structures in the HIV-1 MA shell: The interactions between the Env KS domain and the MA shell could play a critical role during viral entry. Reconstruction of a full Env complex was necessary to predict possible dynamics. Env glycoproteins were therefore incorporated into the MA shell model (Figure 4A). We represented the Env glycoproteins in red with their KS domains in blue to highlight the fact that the KS domains are found within the same plane as the MA trimers (grey). The LLP amphiphilic helixes of the CT domain are found above the MA shell plane and overlap with the myristoyl (myr) groups that help embed MA trimers to the inner membrane of the virion lipid bilayer (Figure 4B). The overall footprint of KS trimers within the plane of the MA shell is 5.64 nm wide and could potentially move within MA shell interlunar gaps. To explore this possibility, we measured distances between adjacent MA lunes across interlunar gaps. From the centre-of-mass of edge MA trimers across the interlunar gap to the centre-of-mass of adjacent edge MA trimers, the average distance was 8.3 nm. The translunar gap could play a key role during viral entry by providing the KS domain of Env glycoproteins the ability to move within the plane of the viral membrane. Randomly incorporated Env glycoproteins significantly impacted interlunar gaps that ranged from 25 to 72.5 nm in length (Figure 4C). Interestingly, model observations predict that the KS domain of Env glycoproteins can get as close as 5.47 nm from an MA trimer centre-of-mass to a KS trimer centre-of-mass (Figure 5C). The three LLP amphiphilic helices of the CT domain do not interfere with the interaction as the helices stay within the inner lipid membrane (Figure 5B). LLP amphiphilic helices are thought to create a plate that anchors Env glycoproteins to the inner HIV-1 lipid membrane but do not restrict the movement of the Env glycoproteins within the plane of the membrane. The movement would be guided by the MA shell interlunar gaps (Figure 5A). A previously conceived hypothesis [4] that Env glycoproteins could occupy the central pore of hexameric MA structures is unlikely. This arrangement generates pores of 3.35 nm in diameter (Figure 5D), which are too narrow to fit the 5.64 nm KS footprint (Figure 3C, blue area).

An updated model of viral entry: In this study, we present an improved molecular model that matches the current understanding of HIV-1 entry. The conceptual model summarizes our findings and predictions made by the in silico model in terms of viral maturation and fusion. In previous models, there was no explanation for how Env glycoproteins were able to cluster to allow fusion to occur. In our novel model, Env glycoproteins first move within the plane of the HIV-1 membrane within the interlunar gaps (more simply “Slide”), then converge toward the fusion hub, forming a cluster of Env glycoproteins, which orients the MA shell to the host membrane to facilitate fusion (Figure 6). In summary, the sequence of Env glycoprotein activity that precedes viral fusion is Slide-Cluster-Fuse. Another advantage of our model is that from the point of membrane fusion, a multi-lune MA shell provides the flexibility needed for the viral particle to change shape and transition to a flattened configuration (Figure 6).

## 4. Discussion

Currently, there is no established role for MA during viral entry. One important function of MA is to recruit the Gag polyprotein to the inner cell membrane during viral assembly. MA membrane targeting occurs through its N-terminal myristoyl groups and the presence of basic residues that interact with acidic phospholipids in the membrane [30,31]. In addition to this role, Env glycoprotein incorporation into the cell membrane is facilitated by MA, as mutations that target MA trimer formation affect Env glycoprotein incorporation [32,33]. The incorporation of Env glycoproteins does not significantly affect the overall number of MA trimers per viral particle. Our calculations show that there is a one-to-one relationship, meaning that for every Env glycoprotein incorporated, there will be one less MA trimer in the final particle. Our updated models generate particles with 3478 MA protein units. In this manuscript, we postulate that the superstructure of the MA shell plays a critical role in facilitating viral entry. This process involves the restriction of Env CT domains to only move within the “trenches” of interlunar gaps not occupied by the MA shell (Figure 5). Evidence supporting this type of arrangement, with Env glycoproteins lined up along an arc along the spherical plane, has been observed in cryoelectron microscopy tomograms of wild-type HIV-1 particles [2].

Characterization of the MA shell using advanced microscopic technologies has proven challenging as electron density patterns of HIV-1 structure are complex and do not follow simple repeated patterns. Some studies have suggested that the MA shell is a uniform hexameric sphere with inexplicable and irregular defects dispersed along both the mature and immature Gag lattice, thus allowing for the spherical curvature [34,35]. Other models postulate that the recruitment of Gag proteins as dimers during lattice formation could result in incomplete or partial lattice edges that allow for spherical shapes [6]. Modelling of incomplete lattice solutions demonstrates the challenge of such postulates (Appendix A). However, electron tomography studies have shown that MA trimers form continuous branches of connected trimers, consistent with the core branches found in our six-lune models [36]. Additionally, the formation of amorphous lunes in our improved model may help explain the irregular defects and gaps observed but unexplained in tomograms. Reports by some research groups observed the traditional hexagonal arrangements [36]; however, these microscopic techniques have been developed to specifically detect patterns such as hexamers and pentamers. Hexagonal pattern observation might be evidence of the biased nature of this approach and the identification of artifacts on semi-flat membranes of misformed particles that are able to accommodate these geometries. The field relies on polyprotein expression systems where transformed bacteria or transfected mammalian cells overexpress polyproteins for HIV-1 lattice investigations. These reductionist approaches could lead to structural models that mislead our understanding of the MA ultrastructure.

Bending a flat two-dimensional Gag hexagonal lattice into a sphere is not thermodynamically favourable as the necessary distortions in intermolecular distances coupled with overlapping protein volumes are simply not possible (Figure 1 and Appendix A). The incorporation of periodic pentagons could solve the geometrical discord. In fact, long-terminal repeat retrotransposons, which are thought to be the evolutionary ancestors to retroviruses such as HIV-1, display uniform distribution of pentagons [37]. Unfortunately, pentagon arrangements for Gag or MA trimers have not been observed. This is additional evidence that instead of visualizing spherical or ellipsoidal viral particles, observed tomographic data may be capturing artifacts, artificially flattened viral particle membranes that may arise during experimental manipulation. Importantly, we recommend that complete cryo-EM electron density volumes are used to report Gag lattices and MA shells, rather than illustrations in which triangles or hexagons are used to depict the raw data (Appendix A). It is difficult for the community to assess the inconsistencies of such models when triangles overlap or are conveniently separated in support of the hexagon hypothesis. We have made our open-source modelling software available online and invite HIV structural biologists to openly collaborate to resolve this outstanding challenge.

All-atom viral simulation models are an instrumental approach to help us decipher components of the viral life cycle [14,38]. Our work highlights MA as a potential key player in HIV-1 viral entry by regulating the movement of Env glycoproteins on the surface of HIV-1 viral particles. We predict that Env glycoproteins do not inhabit the central pore of hexameric structures but rather slide within the interlunar gaps to cluster at the fusion hub, a sequence that we term Slide-Cluster-Fuse (Figure 6). Conceptually, the process of HIV-1 formation involves the random incorporation of Env glycoproteins into the MA shell, with each Env glycoprotein insertion as a nucleation point for the creation of an interlunar gap (Figure 2D). We assembled a full domain model of the Env glycoprotein, including a model of the CT domain using multiple validated prediction tools, which, to our knowledge, has never been demonstrated (Figure 3C). If the hexameric MA configuration was possible, it would generate MA pore sizes of 3.35 nm. The large KS trimeric domain of Env glycoproteins could not fit in this space, as it would require 5.64 nm in our validated prediction model. Our novel Env glycoprotein allows us to visualize the putative footprint of this protein and how it might affect its function. The unusually large CT domain of the Env glycoprotein occupies 25.71 nm. Importantly, the KS domain is too large to fit within the pore that forms between six adjacent MA trimers or hexagon pore. Building the all-atom model of the Env glycoprotein was conducted by using several available structures from the Protein Data Bank and Electron Microscopy Data Bank. Previously, we carefully considered using the NMR structures determined by Piai et al. [26] (PDB ID: 6UJU and 6UJV). Unfortunately, their models omitted amino acids 717-FQTHLPTPRGPDRPEGIEEEGG-738, which constitutes a 51.2% deletion of a central region of the KS domain. In addition to this deletion, the remaining LLP domain only included 49 residues (739–788), excluding 67 residues (789–856). These deletions disconnected the KS and LLP domains making it difficult to assess whether the distances between the TM and the LLP domains were representative of the native conformation. Since determining the volume occupied by a complete KS domain as accurately as possible was a critical aim of our research, we opted to use PDB entries that were more comprehensive. Our in silico structure prediction was constructed using the best available tools to fill this knowledge gap. We obtained the transmembrane domain from a different Protein Data Bank entry (PDB ID: 6E8W) and the full LLP domain from another publication (PDB ID: 5VWL).

All-atom models provide a perspective that takes into account the distance, orientation, and interaction of structural components of viruses [14,38]. This approach allows us to clearly visualize the mechanics of complex viral life cycle processes. Some clear questions regarding the HIV-1 budding and viral entry are unresolved. One critical question is whether the Gag lattice forms a six-lune hosohedron similar to our MA shell superstructure, which could allow for more flexible membrane properties and better interaction with the KS domain of Env glycoproteins. Another process that needs to be elucidated is the molecular interactions that drive the clustering of Env glycoproteins, which has been demonstrated as requisite for viral fusion. Future molecular dynamics and observational studies will be critical to answering these questions. We hope that our hypothetical proposal will be improved upon by observational data in future studies. We are currently developing imaging technology to help us visualize inner membrane structures in small particles such as viruses and exosomes. Our techniques will aim to capture native structural conformations that will help us visualize the relative arrangements of MA trimers on curve surfaces.

By building an atomic model of the Env glycoproteins and integrating this model into the six-lune hosohedron, our study highlights the essential interaction between MA trimers and Env glycoproteins prior to HIV-1 viral entry. This study is an example of how various scientific disciplines with unique perspectives can accelerate the discovery of complex biomolecular processes, such as the gaps that still exist for the life cycle of HIV-1.

## Figures and Tables

**Figure 1 viruses-13-01515-f001:**
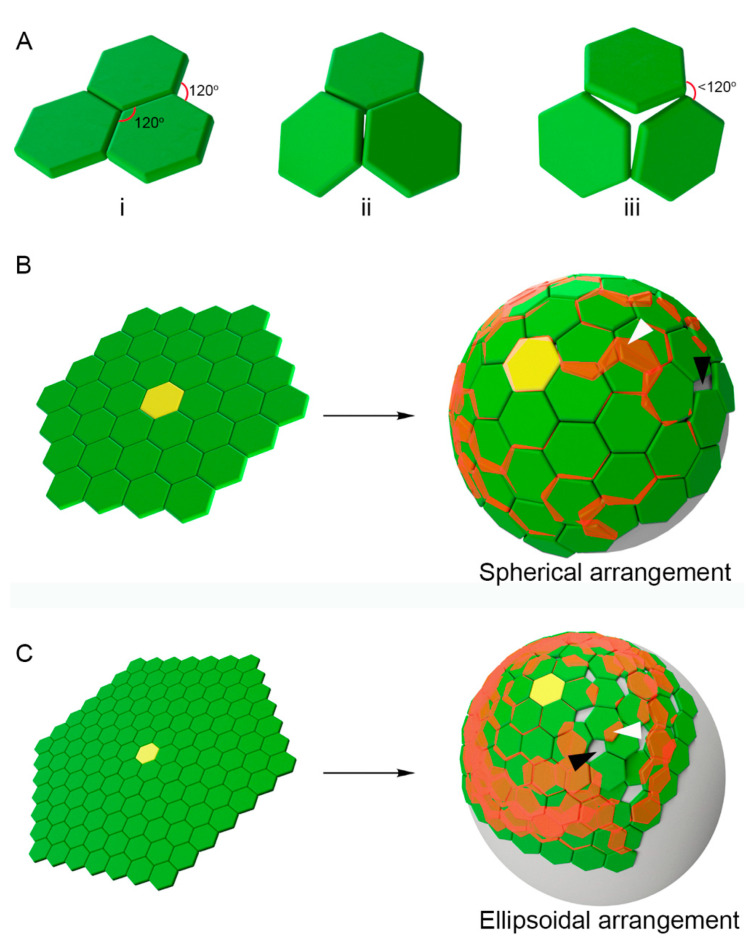
Challenges of the tile-based modelling of HIV-1 matrix (MA) shells. Each *hexagon represents 6 MA trimers.* (**A**) Three tiles arranged on the surface with all angles at 120° transition to a curve surface (**Ai**–**iii**). External angles become less than 120° if tiles are not flat. (**Aiii**) The angle for adjacent MA hexamers to join is decreased (<120°). Thus, the configuration of MA hexamers forming a spherical/ellipsoidal shell is implausible. (**B**) A yellow tile represents the origin MA hexamer from which adjacent MA hexamers connect and expand outwards. In a flat surface, MA hexamers can be added without disrupting geometric permissiveness. If the surface is bent into a spherical arrangement, overlap of tiles (MA hexamers) takes place. Overlapping regions are highlighted in red (white arrowheads). Black arrowhead indicates empty space. (**C**) Similar representation as (**B**) following an ellipsoidal curvature.

**Figure 2 viruses-13-01515-f002:**
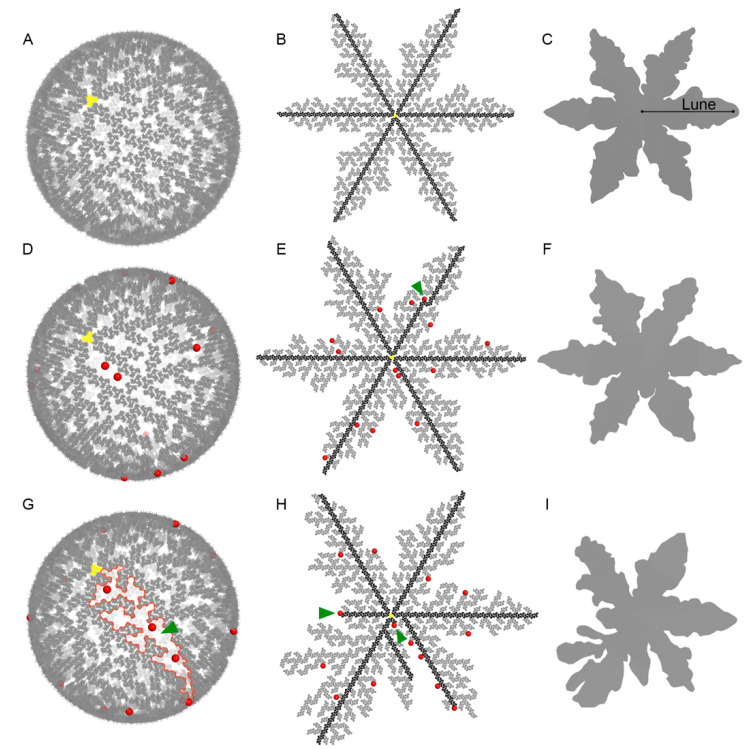
HIV-1 MA shell model. (**A**) The origin matrix trimer (MA) is depicted in yellow. Additional MA trimers extend from the initial yellow trimer with no overlaps between trimers or gaps where MA trimers could be fitted. (**B**) Flat configuration from (**A**) with core trimer branches highlighted in black. Core trimers generate 6 lunes. (**C**) Silhouette of (**B**) with one of the six lunes labelled. (**D**) A total of 14 red dots highlight the location where MA trimer propagation was interrupted to represent the incorporation of envelope glycoprotein complex (gp120/gp41, or Env). (**E**) Flat arrangement of (**D**). Green arrowhead represents a termination that occurred at a core MA trimer. (**F**) Silhouette of (**E**). (**G**) Similar to (**D**) but with shell interruptions that occurred in early core trimers. Interlunar space is outlined in red (green arrowhead). (**H**) Flat arrangement of (**G**). Green arrows indicate locations where core lune branches were interrupted. (**I**) Silhouette MA shell of (**H**).

**Figure 3 viruses-13-01515-f003:**
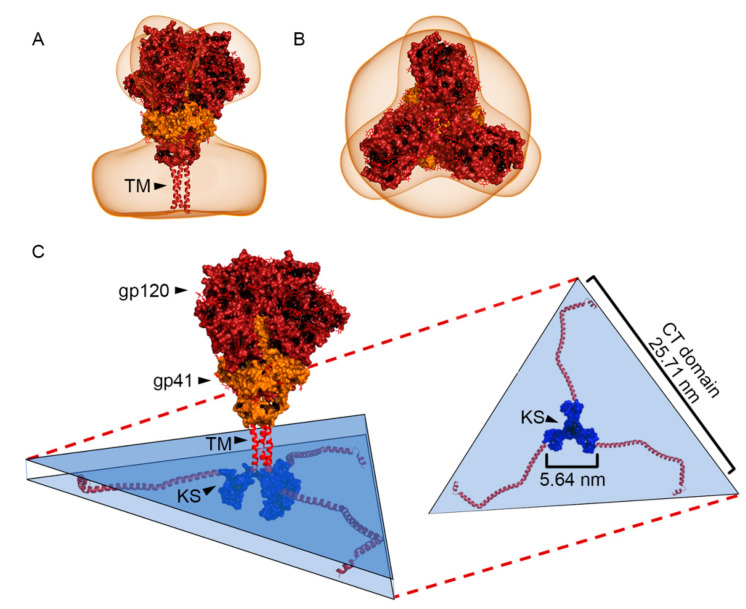
Three-dimensional representation of Envelope glycoprotein complex (Env, gp120/gp41). (**A**) Side view of gp120 and part of gp41 fitted to an electron tomogram (brown outline). (**B**) Top view of (**A**). (**C**) Entire Env complex comprising the transmembrane domain (TM) in red, the Kennedy sequence domain (KS) in blue, and the LLP amphiphilic helices (red helices branching out from KS). A blue triangular prism highlights a 3D volume that encloses the CT domain. Top-down view of the triangular prism shows the CT domain and its dimensions.

**Figure 4 viruses-13-01515-f004:**
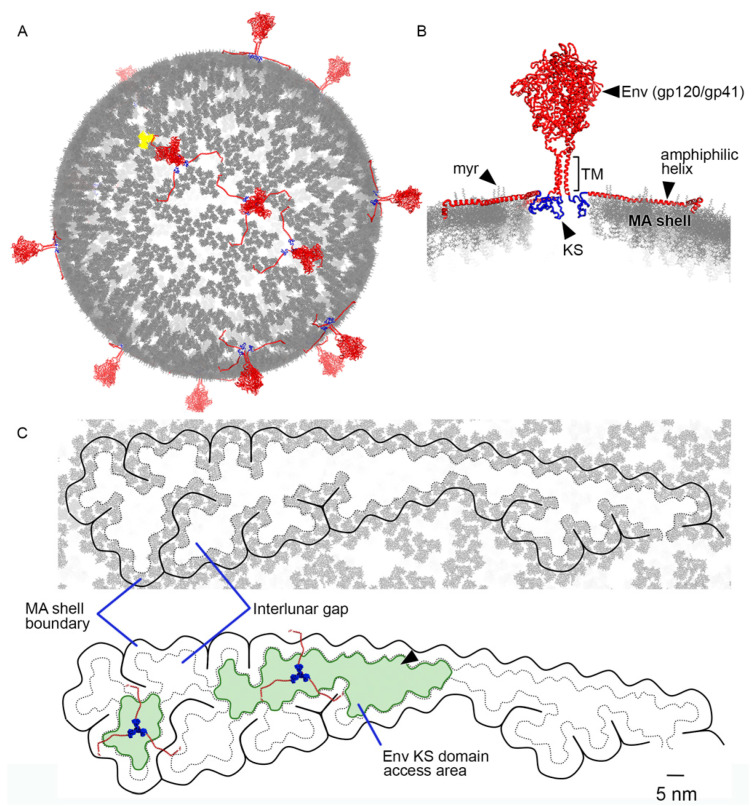
HIV-1 MA shell model and its interaction with Env glycoproteins. (**A**) Three-dimensional representation of MA shell (MA origin trimer, yellow) with Env glycoprotein complexes (Env, red) embedded in interlunar spaces. (**B**) Cross-sectional view of the MA shell and one Env glycoprotein highlighting the relative position of a KS domain to the MA shell. MA myristoyl groups (myr) are shown. Env transmembrane domain (TM), cytoplasmic domain LLP amphiphilic helixes associated with the inner membrane are also shown (black arrowheads). (**C**) Interlunar gaps where Env KS domains can fit (depicted in green) and may move within the plane of the viral membrane. The MA shell boundary and the interlunar space are indicated in the image. Large interlunar spaces that take place when MA core branches are interrupted by the incorporation of Env glycoproteins during viral formation (black arrowhead).

**Figure 5 viruses-13-01515-f005:**
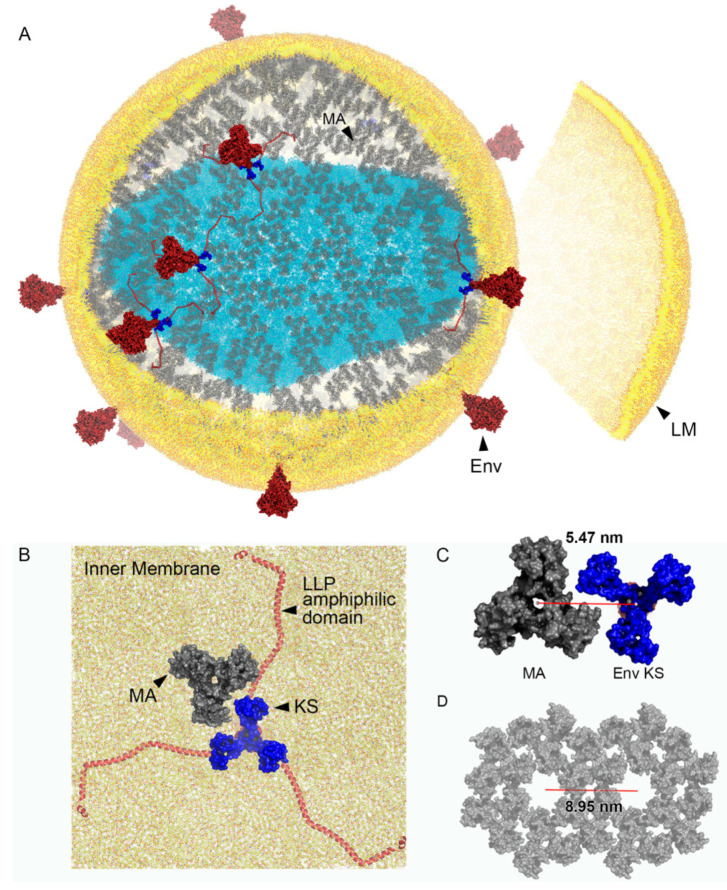
HIV-1 model including the MA shell, Env glycoproteins, and a lipid membrane. (**A**) Three-dimensional model of an HIV-1 virion showing a lipid membrane (LM, yellow), MA shell (grey), Env glycoprotein complexes (Env, red; KS domain in dark blue), and capsid (light blue). (**B**) View from inside of virion focusing on a MA trimer next to an Env glycoprotein (**C**) 5.47 nm is the minimum distance between the centre-of-mass of an MA trimer and the centre-of-mass of a KS domain. The KS domain is 5.64 nm wide, see Figure 3C. (**D**) Side-by-side flat MA hexamers separated at a distance of 8.95 nm (red line). Central space within hexamers has a diameter of 3.35 nm.

**Figure 6 viruses-13-01515-f006:**
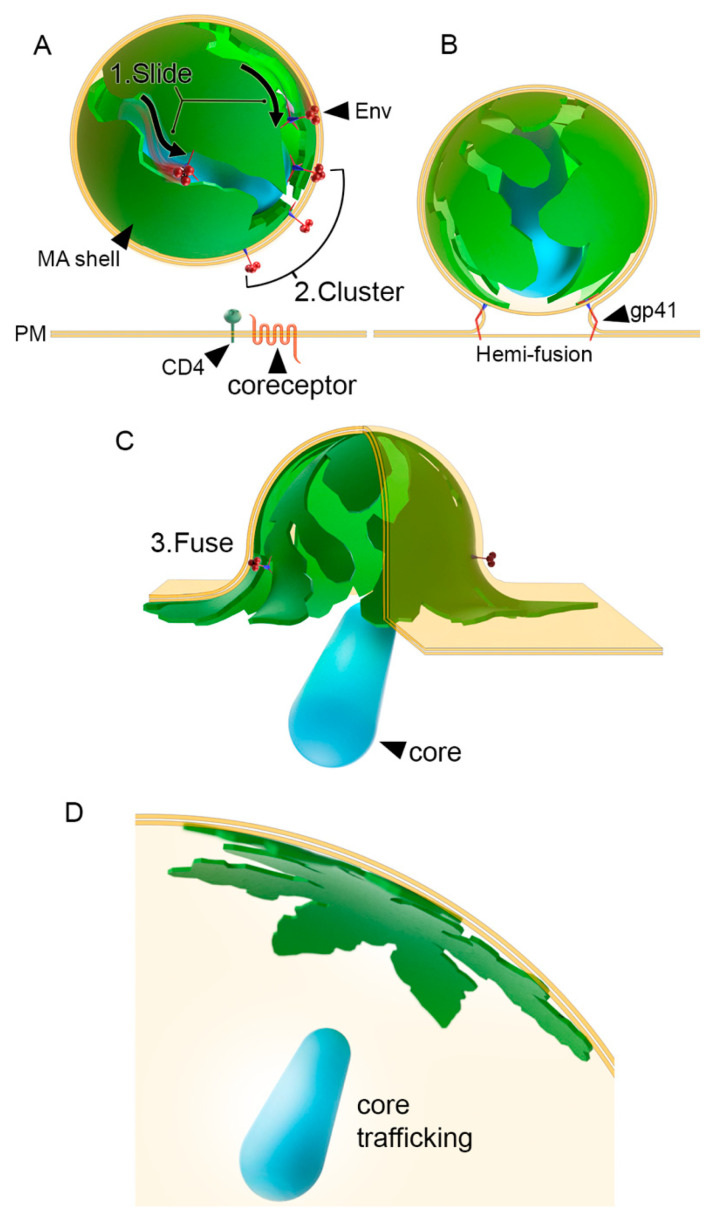
Conceptual model of HIV-1 viral entry. (**A**) During viral maturation, Env glycoproteins slide and cluster at the fusion hub. (**B**) Clustered Env glycoproteins facilitate the interaction of multiple Env glycoproteins with CD4 receptors and coreceptors, inducing a hemi-fusion state. (**C**) Viral fusion completes the Slide-Cluster-Fuse sequence (steps labelled 1, 2, and 3), leading to (**D**) a flattened MA shell (green) and trafficking of the capsid core to the nucleus.

## Data Availability

Not applicable.

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
