# Peer review of "Challenging the Existing Model of the Hexameric HIV-1 Gag Lattice and MA Shell Superstructure: Implications for Viral Entry"

_viruses, 2021, doi:10.3390/v13081515_

Round 1

Reviewer 1 Report

The manuscript by Santos et al. describes a model to explain the Gag lattice of HIV-1 and the structure of the envelope-matrix interface in the viral particle. Authors propose a model based on a 6-lune hosohedral structure. Their article addresses a potential limitation of a previous model that authors described in 2019 (#14 in references section).

Although, I still think that authors’ conclusions are mostly hypothetical, the manuscript might be useful for researchers working on HIV structure and virion maturation.

The manuscript has been improved from the last version I reviewed, however, there are still issues that require attention:

  • Abstract, highlighted lines 19-20 appear duplicated in the following lines. Besides, in line 21, it should read ‘hosohedral structure’
  • Introduction, first paragraph, lines 35-40. Very often, Env designation is not precise. In my opinion, it is better to refer to Env (gp120/gp41) as the mature envelope glycoprotein, composed of subunits gp120 and gp41. It derives from cleavage of gp160 as Env precursor.  This concept is implicit in the title of one of the sections of the manuscript (at line 136, p. 4). However, in the introduction, authors refer to Env proteins as two different glycoproteins, and sentences might not be accurate. It is fine to assume that gp120 and gp41 are different entities, but in that case I would not use the term Env (just ‘envelope glycoproteins’). Please revise the text accordingly.
  • Page 1, line 36 > envelope
  • Page 2, lines 61-62: > interaction of gp120 trimers; > CXCR4
  • Although I understand the experimental limitations, authors should provide in the discussion some ideas about how to test the validity of their model.

Reviewer 2 Report

The Santos et al. Viruses manuscript, "Challenging the existing model of the hexameric HIV-1 Gag lattice and MA shell superstructure: Implications for viral entry" is an extension of previous work that proposed a 6-lune hosohedral model for matrix (MA) protein interactions in HIV-1 particles. The current manuscript endeavors to refine the previous model by incorporating HIV-1 envelope protein trimers into the calculations.

On the positive side, the calculations and models proposed offer an interesting perspective that may stimulate readers into considering alternatives for how MA may organize in HIV-1 particles. On the negative side, the authors have not done a good job in explaining how hexamer models may or may not be accommodated by quasi-equivalence and/or formation of incomplete or partial lattices. Perhaps a bigger problem is that the authors have not taken into account new data concerning the structure of the Env CT (Piai et al, Nature Communications, 2020) or the tomographic observation of incomplete hexameric lattices in both mature and immature HIV-1 particles (Qu et al., bioRxiv, 2020). While the latter report is a preprint that has not been certified by peer review, the data are so relevant to the current submission that it would seem essential to address the observations. Some suggestions for improving the manuscript are as follows:

  1. The authors should clarify why quasi-equivalence or lattices with breaks can not mitigate the energy constraints imposed by hexagonal MA lattices.
  2. The results in the Qu et al report ought to be addressed directly.
  3. The observations of Piai should be incorporated into the calculated models.
  4. It would be particularly worthwhile for the authors to offer specific predictions as to how their models might be tested. For instance, does Env incorporation change the prediction of how many MA proteins are included in virus particles? Or what MA residues (if any) would be expected to participate in Env incorporation and why? Or how are mature and immature MA lattices expected to differ?
  5. Explain the limitations of assuming that lattices were seeded by a single MA trimer, and why seeding by a single trimer is or is not physiologically relevant.

6. Delete Figure 6, which does not add to the results of the manuscript. 

Round 2

Reviewer 2 Report

The revised version of the Santos et al. manuscript appears improved from the initial manuscript and is likely to attract the attention of readers interested in the structure of HIV-1, although several responses to previous concerns were somewhat perfunctory. These include the following:

  1. The Piai Env CT structure was dismissed in favor of the Murphy et al. CT structure. The concern here is that the Murphy et al. structure excludes Env residues 707-751, and shows an elongated, potentially unnatural conformation, due to the absence of the Env transmembrane (TM) domain. While it is true that the Piai structure includes the Env TM and only CT residues 739-788, the structure indicates that the CT appears to wrap around ends of the TM helices in a much tighter conformation than the extended Murphy et al. structure.
  2. The justification for assuming that MA lattices are seeded by a single trimer could use bolstering. The first argument, that there are roughly 1000 MA trimers in a single particle, does not really address the possibility of multiple lattice seeds. The second and third arguments, that "single Gag trimers have more degrees of freedom to move at higher speeds than bulkier pre-formed lattices" and that "adjacent arcs are unlikely to fuse" presumes that the final structure represents a single lattice, rather than multiple, small unfused lattices.

3. The dismissal of using gaps to build MA lattices is directly connected to the assumption that MA lattices are seeded by single trimers. In the new Supplemental Figure 1A, there were 16 hexamers that were free of overlap, 11 of which formed a single patch. This presumably implies that patches of up to 11 hexamers are permitted; and that up to 14-15 such patches could be separately seeded (and unfused with each other) in a single virus particle. 

This manuscript is a resubmission of an earlier submission. The following is a list of the peer review reports and author responses from that submission.

Round 1

Reviewer 1 Report

The manuscript by Santos et al. describes a model to explain the Gag lattice of HIV-1 and the structure of the envelope-matrix interface in the viral particle. Authors propose a model based on a 6-lune hosohedron structure. Hosohendrons resemble a snowflake structure and could explain movement of Env within the surface of the virion through interlunar spaces. Their article addresses a potential limitation of a previous model that authors described in 2019 (#13 in references section). This is nicely written article with excellent illustrations. However, I think that authors’ conclusions are mostly hypothetical and do not rely on experimental data.  Although plausible, I think that their proposal needs more solid experimental evidence in order to supersede current models. Still, the manuscript could be accepted as a hypothetical proposal.

Comments:

.- Abstract: Authors refer to a “6-lune hosohedra model” and “lune hosohedra” (it sounds better to me: “hosohedral or hosohedron”?). I think that this is not a common term for biologists and a reference to its similarity to a snowflake structure should be included in the abstract for better clarification.

.- All Figures contain labels (e.g. Fig. 1, Fig. 2, Fig. 3) that should be eliminated in the published images.

.- Reading the manuscript it is not clear to me which model in Figure 2 is the proposed one. Although authors refer to a 6-lune structure (Fig. 2A-C), the stronger compatibility would be with models G-I, which consider the presence of Env molecules.

.- Considering that Koch snowflakes are classical fractal structures, would be a fractal structure compatible with the formation of a Gag lattice? Would the presence of Env structures affect this fractal formation and hence, the lattice?

Reviewer 2 Report

Santos et al. refine a previously published model of HIV structure by including a model of the Env structure into their theoretical framework.

While the original model is interesting, it does not predict the data from real viral particles published especially by the Briggs lab and obtained by Cryo-electron tomography in their native condition (https://doi.org/10.1073/pnas.0903535106, https://journals.plos.org/plospathogens/article?id=10.1371/journal.ppat.1001215 and especially a recent paper: https://doi.org/10.1073/pnas.2020054118). Unfortunately the author largely miss to discuss this data. This reviewer also does not understand an argument (lines 354-356) why cryo-tomography of virus particles would represent 2D lattices.